# Adipophilin expression is an independent marker for poor prognosis of patients with triple-negative breast cancer: An immunohistochemical study

**Katsuhiro Yoshikawa**[1,2], **Mitsuaki Ishida**[1]*, **Hirotsugu Yanai**[2], **Koji Tsuta**[1], **Mitsugu Sekimoto**[2], **Tomoharu Sugie**[2]

**1** Department of Pathology and Clinical Laboratory, Kansai Medical University, Osaka, Japan, **2** Department of Surgery, Kansai Medical University, Osaka, Japan

* ishidamt@hirakata.kmu.ac.jp

**Data Availability Statement:** All relevant data are within the manuscript and its Supporting Information files.

## Abstract

Adipophilin is a lipid droplet-associated protein whose expression can act as a prognostic marker for specific cancers. Using immunohistochemical staining and tissue microarrays, we assayed the expression of adipophilin in 61 patients with triple-negative breast cancer (TNBC) who underwent surgery from January 2006–December 2018. Relapse-free survival (RFS) and its risk factors were analyzed based on adipophilin expression. Fourteen (23.0%) patients expressed adipophilin. As compared to the adipophilin-negative TNBC patients, adipophilin-positive patients exhibited poor RFS ($p = 0.032$). Among the TNBC patients with a high Ki-67 labeling index, patients negative for adipophilin exhibited better RFS than patients positive for adipophilin ($p = 0.032$). Moreover, among patients who did not undergo adjuvant chemotherapy, patients negative for adipophilin expression exhibited better RFS than adipophilin-positive patients ($p = 0.080$). Multivariate analysis showed that adipophilin expression correlated with a high rate of relapse (hazard ratio, 4.89; 95% confidence interval, 1.04–23.0; $p = 0.044$). Taken together, these results indicate that adipophilin is a novel marker for the poor prognosis of patients with TNBC.

## Introduction

Triple-negative breast cancer (TNBC), a high-grade breast cancer with poor prognosis, is characterized by the lack of estrogen and progesterone receptors and human epidermal growth factor receptor 2 (HER2) [1, 2]. The Nottingham Prognostic Index, lymph node status, tumor size, and Ki-67 labeling index (LI) are useful prognostic markers, albeit non-specific for TNBC [3, 4]. The risk of cancer recurrence in patients with hormone receptor-positive breast cancer can be predicted using multigene assays, such as Oncotype DX®. This helps determine which patients should receive chemotherapy [5–7]. However, these tests cannot predict the prognosis of TNBC. Most TNBC patients undergo chemotherapy owing to the lack of a method to

**Funding:** The authors received no specific funding for this work.

**Competing interests:** The authors have declared that no competing interests exist.

evaluate the risk of recurrence. Therefore, there is an urgent need to identify novel biomarkers for TNBC.

Adipophilin, also known as perilipin 2, is a lipid-regulating protein of the perilipin/adipophilin/tail interacting protein of 47 kDa (PAT) family that coats the surfaces of cytoplasmic lipid droplets [8, 9]. Lipids are essential for tumor cell proliferation [10]. PAT family proteins are expressed in various types of cancer cells [11–16]. Recent studies have demonstrated the correlation between adipophilin expression and poor prognosis of some types of cancers, including lung adenocarcinoma [14] and pancreatic ductal adenocarcinoma [16]. Expression of adipophilin has been reported in breast cancer tissues [17]; however, the prognostic value of adipophilin expression in TNBC remains to be elucidated. Thus, the aim of this study is to determine the correlation between adipophilin expression and prognosis of patients with TNBC.

## Materials & methods

### Patient selection

We selected 165 consecutive patients with TNBC who underwent surgical resection at the Department of Surgery of the Kansai Medical University Hospital in January 2006–December 2018. Patients who were administered for neoadjuvant chemotherapy or those who had a special type of invasive carcinoma were excluded from the study; the study cohort comprised 61 TNBC patients (S1 Fig).

This study was conducted in accordance with the principles embodied in the Declaration of Helsinki and the study protocol was approved by the Institutional Review Board of the Kansai Medical University Hospital (Approval #2019041, #2019234). Informed consent was obtained from patients by opt-out methodology owing to the retrospective design of the study, with no risk for the participants. Information regarding this study, such as the inclusion criteria and opportunity to opt out, was provided through the institutional website. (http://www.kmu.ac.jp/hirakata/hospital/2671t800000135f8-att/a1582783385210.pdf).

### Histopathological analysis

Surgically resected specimens were fixed with formalin, sectioned, and stained with hematoxylin and eosin. All histopathological diagnoses were evaluated independently by more than two experienced diagnostic pathologists. We used the AJCC/UICC TNM classification and stage groupings. Histopathological grading was based on the Nottingham histological grade evaluated in the tumor tissue using tissue microarray [18]. We also evaluated the presence of clear cytoplasm, eosinophilic cytoplasm, and prominent nucleoli in the tumor tissue using tissue microarray. Clear or eosinophilic cytoplasm was considered to be present when these histological changes were identified in >5% of the tumour cells, as described earlier [14].

The Ki-67 LI, which was also evaluated in the tissue microarray, was considered high when it was 40% or more, according to a meta-analysis of TNBC patients [19]. The Ki-67 LI of two patients were not evaluated because less than 1,000 carcinoma cells were present in the tissue microarray.

### Tissue microarray

Hematoxylin and eosin-stained slides were used to select the most morphologically representative carcinoma regions. Three tissue cores of 2 mm diameter were punched out from the paraffin-embedded blocks of each patient sample. Tissue cores were arrayed in the recipient paraffin block.

## Immunohistochemistry

Immunohistochemical analyses were performed using an autostainer (Discovery ULTRA System; Roche Diagnostics, Basel, Switzerland) according to the kit instructions. A primary mouse monoclonal antibody was used to detect adipophilin (AP125, Progen Biotechnik, Heidelberg, Germany; diluted at 1:100). For antigen retrieval, tissue sections were autoclaved at 100˚C for 20 minutes in a tris-based buffer, pH 8.5, (Conditioning Solution CC1, Ventana Medical System). Thereafter, the automated protocol steps for immunostaining were followed. In addition, 3,3'-diaminobenzidine (DAB) was used as a colorimetric agent. Human sebaceous glands tissues were used as positive controls for adipophilin immunoreactivity. At least two researchers evaluated the immunohistochemical staining independently. Adipophilin expression was considered positive when the neoplastic cells showed granular and/or globular cytoplasmic expression as reported previously (Granular staining pattern is defined as staining of round smaller size, while globular staining pattern is defined as staining of round to oval larger size.) [14, 16]. The patient was considered adipophilin-positive when one or more cores from the same patient showed positive immunoreactivity as described previously [14, 16]. In order to determine the cut-off value for adipophilin positivity, analysis was performed with positive cut-off values of 10, 20, 30, 40, and 50%.

## Statistical analyses

All analyses were performed using SPSS Statistics 25.0 (IBM, Armonk, NY, USA). Correlation between two groups was determined using Fisher's exact test for categorical variables and Mann–Whitney U test for continuous variables. The rate of relapse-free survival (RFS) was evaluated using Kaplan–Meier analysis; log-rank tests were used to compare between groups. The Cox proportional-hazards model was used to examine the correlation between clinicopathological parameters and survival. $p < 0.05$ was considered statistically significant.

# Results

## Patient characteristics

This study comprised 61 female patients. The median age at the time of initial diagnosis was 58 years (range: 31–93 years). All patients were diagnosed with TNBC based on biopsy results, and all samples were invasive carcinoma, no special type. No discrepancy was found in the pathological diagnosis and molecular subtype between the pre-operative biopsy specimens and operative specimens. The median tumor diameter was 20 mm (range: 2–55 mm). Patients were staged as I (25 patients), IIA (23 patients), IIB (5 patients), IIIA (4 patients), IIIB (3 patients), and IIIC (1 patient). Based on the histopathology, 2, 27, and 32 patients had tumor grades 1, 2, and 3, respectively. The Ki-67 LI was high, low, and not evaluated in 26, 33, and 2 patients, respectively. The observation period was up to 36 months in all patients. Eight (13.1%) patients experienced relapse (all had distant metastasis, and none experienced local recurrence), and five (8.2%) patients died of the disease. Clinical characteristics of all 61 patients can be found online in S1 Table.

## Correlation between adipophilin expression and clinicopathological factors

Table 1 shows the correlation between adipophilin expression and clinicopathological factors in the study cohort. Only cut-off value of 30% was significantly associated with RFS ($p = 0.032$), while cut-off values of 10, 20, 40, and 50% were not significantly associated with RFS ($p = 0.280, 0.072, 0.07,$ and $0.07$, respectively). Therefore, in the present study, the cut-off value was set at 30% for the subsequent analyses. Fourteen patients (23.0%) were adipophilin-

**Table 1.** Correlation between adipophilin expression and clinicopathological factors.

| Factors | ADP-positive (n = 14) | ADP-negative (n = 47) | *P*-value |
|---|---|---|---|
| **Age (years; mean ± standard deviation)** | 66 ± 15 | 65 ± 16 | 0.850 |
| **Menopausal status** | | | |
| Premenopausal | 2 | 7 | 1.000 |
| Postmenopausal | 11 | 40 | |
| Unknown | 1 | 0 | |
| **Body mass index** | 23.7 ± 3.6 | 23.4 ± 3.8 | 0.674 |
| **Tumor size (mm)** | | | |
| ≦20 | 7 | 24 | 1.000 |
| >20 | 7 | 23 | |
| **Pathological stage** | | | |
| I+II | 12 | 41 | 1.000 |
| III | 2 | 6 | |
| **Lymph node status** | | | |
| Positive | 5 | 9 | 0.263 |
| Negative | 6 | 27 | |
| Not tested | 3 | 11 | |
| **Lymphatic invasion** | | | |
| Positive | 13 | 40 | 0.668 |
| Negative | 1 | 7 | |
| **Venous invasion** | | | |
| Positive | 10 | 27 | 0.534 |
| Negative | 4 | 20 | |
| **Nottingham histological grade** | | | |
| 1 + 2 | 5 | 24 | 0.372 |
| 3 | 9 | 23 | |
| **Ki-67 labeling index** | | | |
| High | 11 | 15 | **0.005** |
| Low | 3 | 30 | |
| Not evaluated | 0 | 2 | |
| **Clear cytoplasm** | | | |
| Present | 1 | 5 | 1.000 |
| Absent | 13 | 42 | |
| **Eosinophilic cytoplasm** | | | |
| Present | 1 | 11 | 0.264 |
| Absent | 13 | 36 | |
| **Prominent nucleoli** | | | |
| Present | 4 | 7 | 0.256 |
| Absent | 10 | 40 | |
| **Adjuvant chemotherapy** | | | |
| Performed | 7 | 28 | 0.749 |
| Not performed | 6 | 17 | |
| Undetermined | 1 | 2 | |

positive and 47 (77.0%) were adipophilin-negative. Most cases exhibited a pan-cytoplasmic globular pattern of staining (Fig 1). Adipophilin expression did not correlate with any clinical factors including age, menopausal status, body mass index, or adjuvant chemotherapy. A high

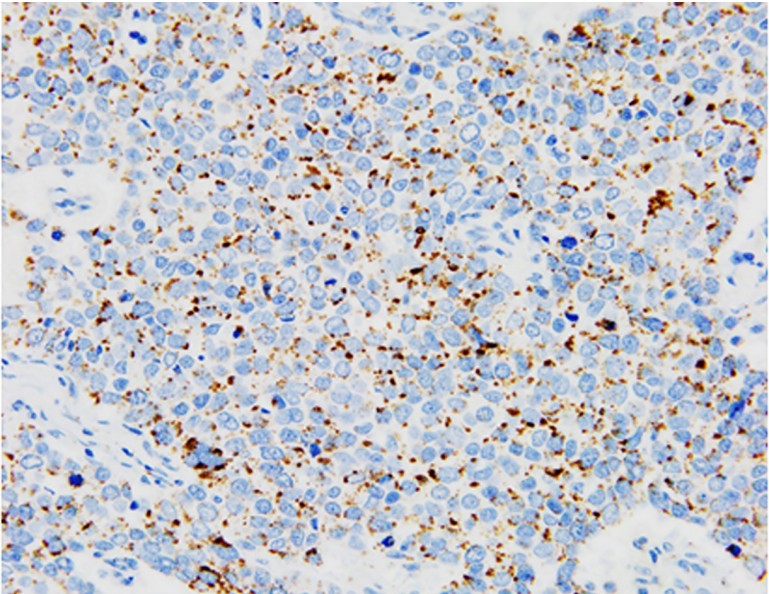

**Fig 1. Representative images of the immunohistochemical staining for adipophilin in triple-negative breast cancer (×200 magnification).**

Ki-67 LI was significantly correlated with adipophilin expression ($p = 0.005$), but not with any other factors including tumor diameter, pathological stage, histological grade, lymphatic invasion, venous invasion, and lymph node status. Moreover, no differences were observed between the morphological features of adipophilin-positive and -negative cases, including the presence of clear cytoplasm, eosinophilic cytoplasm, and prominent nucleoli.

### Correlation between adipophilin expression and postoperative RFS

Fig 2A shows the RFS curves of adipophilin-positive and -negative patients. The frequency of 36-month RFS was 71.4% and 91.5% for adipophilin-positive and -negative patients, respectively. Adipophilin expression significantly correlated with poor RFS ($p = 0.032$) of TNBC patients. Among TNBC patients with a high Ki-67 LI, those negative for adipophilin exhibited better RFS than those positive for adipophilin ($p = 0.032$; Fig 2B). Among the adipophilin-negative patients, there was no difference in RFS between patients with high and low Ki-67 LI ($p = 0.215$). In TNBC patients without adjuvant chemotherapy, adipophilin-negative patients showed a trend for better RFS than adipophilin-positive patients ($p = 0.080$; Fig 2C).

### Prognostic potential of adipophilin expression

Univariate and multivariate analyses were used to determine the effects of clinicopathological factors on RFS (Table 2). Based on the univariate analysis, advanced pathological stage (stage III), lymph node metastasis, and adipophilin expression significantly correlated with poor RFS ($p = 0.036$, $0.025$, and $0.048$, respectively). In contrast, multivariate Cox hazard analyses showed that adipophilin expression was an independent factor for poor prognosis of patients with TNBC (hazard ratio: 4.89; 95% confidence interval; 1.04–23.0; $p = 0.044$). Moreover, adipophilin expression was independent from adjuvant chemotherapy in RFS (hazard ratio: 4.12; 95% confidence interval; 1.02–16.6; $p = 0.047$).

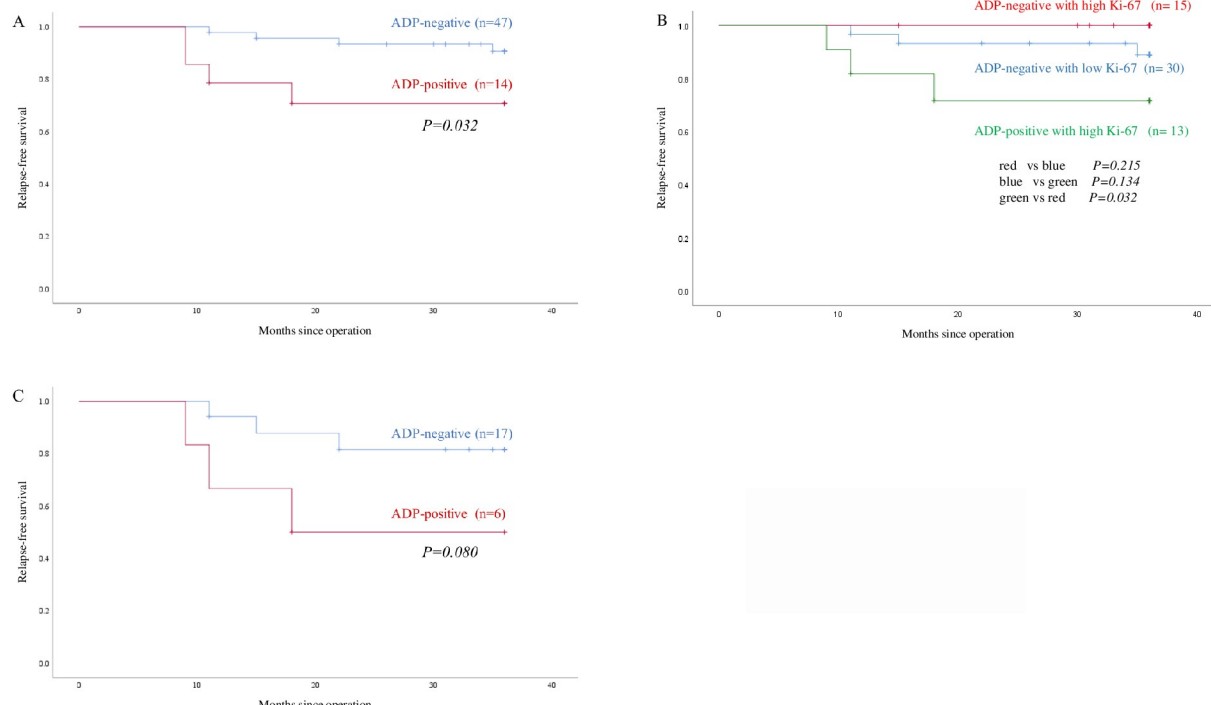

**Fig 2. Kaplan–Meier curves for the relapse-free survival (RFS) of triple-negative breast cancer patients.** (A) RFS curves in adipophilin-positive (red line) and -negative (blue line) patients. (B) RFS curves of adipophilin-negative patients with a low Ki-67 labeling index (LI; blue line), adipophilin-negative patients with a high Ki-67 LI (red line), and adipophilin-positive patients with a high Ki-67 LI (green line). (C) RFS curves in TNBC patients (adipophilin-positive [red line] and -negative [blue line]) without adjuvant chemotherapy.

**Table 2. Univariate and multivariate analysis of relapse-free survival.**

| Factor | | Univariate analysis | | | Multivariate analysis | |
|---|---|---|---|---|---|---|
| | HR | 95% CI | *P*-value | HR | 95% CI | *P*-value |
| **Tumor size (mm)** | | | | | | |
| 20 < vs ≦ 20 | 0.98 | 0.25–3.92 | 0.981 | | | |
| **Pathological stage** | | | | | | |
| III vs I + II | 4.64 | 1.11–19.5 | *0.036* | 2.47 | 0.44–14.0 | 0.306 |
| **Lymph node status** | | | | | | |
| positive vs negative | 6.56 | 1.27–33.8 | *0.025* | 3.65 | 0.55–24.5 | 0.182 |
| **Lymphatic invasion** | | | | | | |
| positive vs negative | 24.5 | $0.02–3.3×10^5$ | 0.510 | | | |
| **Venous invasion** | | | | | | |
| positive vs negative | 4.46 | 0.55–36.2 | 0.162 | | | |
| **Nottingham histological grade** | | | | | | |
| 3 vs. 1+2 | 1.65 | 0.39–6.92 | 0.493 | | | |
| **Ki-67 labeling index (LI)** | | | | | | |
| high vs. low | 0.973 | 0.22–4.35 | 0.971 | | | |
| **Adjuvant chemotherapy** | | | | | | |
| not perform vs. perform | 4.72 | 0.95–23.5 | 0.058 | | | |
| **Adipophilin expression** | | | | | | |
| positive vs. negative | 4.05 | 1.01–16.3 | *0.048* | 4.89 | 1.04–23.0 | *0.044* |

## Discussion

In this study, we examined the clinicopathological significance of adipophilin expression in TNBC patients. We demonstrated the following findings: expression of adipophilin was an independent factor for determining the prognosis of patients with TNBC using multivariate analysis; adipophilin-negative patients with a high Ki-67 LI had a significantly better prognosis compared to that of adipophilin-positive patients with a high Ki-67 LI; and among TNBC patients without adjuvant chemotherapy, those negative for adipophilin exhibited a better prognosis compared to those positive for adipophilin. To the best of our knowledge, this is the first study addressing the prognostic significance of adipophilin expression in TNBC, although a previous study reported that adipophilin expression was frequently associated with TNBC and HER2 subtypes compared to that in luminal subtypes and higher Ki-67 LI [17].

Many studies have attempted to identify prognostic factors in patients with TNBC [3]. The Nottingham Prognostic Index, lymph node status, tumor size, pathological stage, and Ki-67 LI are known prognostic markers [3, 4]. In this study, univariate analysis for RFS showed that pathological stage (III vs I+II) and lymph node status were significant prognostic factors ($p = 0.036$ and $0.025$, respectively), but Nottingham histological grade and Ki-67 LI were not. Multivariate analysis showed that pathological stage and lymph node status were not associated with RFS whereas adipophilin expression was a prognostic factor for poor RFS in patients with TNBC ($p = 0.044$). These results indicate that adipophilin expression is a useful prognostic marker for RFS in TNBC patients. Moreover, the presence of clear cytoplasm, eosinophilic cytoplasm, and prominent nucleoli was not associated with adipophilin expression in TNBC. Some previous reports showed that the presence of clear cytoplasm was not associated with adipophilin expression in lung adenocarcinoma and pancreatic ductal adenocarcinoma [14, 16].

Interestingly, adipophilin expression was a significant marker for poor prognosis in patients with a high Ki-67 ($p = 0.032$); in contrast, Ki-67 LI (high or low) did not correlate with RFS in adipophilin-negative patients ($p = 0.215$). These results indicate that adipophilin is a superior prognostic marker in patients with TNBC, and analysis of adipophilin might better identify patients with a favorable prognosis when combined with Ki-67. Moreover, adipophilin-negative patients without adjuvant chemotherapy showed better prognosis. To the best of our knowledge, there are no established markers for determining whether adjuvant chemotherapy should be administered to patients. Therefore, adipophilin-negative patients may be subjected to de-escalation treatment for TNBC. More prospective clinical studies are required to validate this hypothesis.

Consistent with the results in this study, previous studies demonstrated that adipophilin expression was a factor for poor prognosis for several cancers [14, 16]. The mechanism involved in adipophilin expression in these cancers remains unclear. Adipophilin expression is associated with upregulated lipid synthesis in neoplastic cells [14, 16]. Aerobic glycolysis is the primary energy-generating pathway in cancer cells; this is known as the Warburg effect [10] that results in increased concentrations of intracellular lipids. Increased cancer cell proliferation requires large amounts of lipids to produce cell membranes [20, 21]. Lipid metabolism is associated with TNBC growth [22], and adipophilin expression correlates with higher Ki-67 [17]. Moreover, hypoxia drives the activation of adipophilin in breast cancer cell lines [23], and highly proliferative breast cancer cells thrive in hypoxic conditions [24, 25]. Thus, adipophilin expression in TNBC reflects upregulated lipid metabolism that correlates with a higher proliferative capacity of cancer cells in hypoxic tumor microenvironments [23].

It is important to note the limitations associated with this study. First, this was a retrospective single-institution study with a small sample size that could have led to selection bias.

Second, because TMA cores of 2 mm in diameter were used to determine adipophilin expression in patients, there could be heterogeneous expression of adipophilin in the cancer tissues even though we selected regions that were most representative (morphologically) of cancer. Further studies based on adipophilin immunostaining using whole tissue sections are required to validate our results. Moreover, differential adipophilin expression in operative and biopsy specimens needs to be evaluated. Third, since chemotherapy affects the expression of adipophilin [14], this study excluded patients who were administered neoadjuvant chemotherapy. Further experiments are needed to clarify whether adipophilin is an independent prognostic factor for TNBC patients with and without neoadjuvant chemotherapy. Fourth, we used a follow-up period of more than 36 months based on published data that showed the risk of recurrence in TNBC patients' peaks within 3 years [2, 26]. However, recurrence at a later period in this cohort is possible, and the significance of adipophilin expression in RFS might be different during this period. Finally, this study focused on the expression of adipophilin in TNBC patients. Because adipophilin expression differs among molecular subtypes of breast cancer [17], its prognostic value might be different in patients with the luminal and HER2 subtypes. Thus, further analyses are needed to investigate the prognostic value of adipophilin expression in patients with breast cancers in addition to TNBC.

In conclusion, this study demonstrates that adipophilin expression is an independent factor for poor prognosis of patients with TNBC. Additional studies are needed to elucidate the molecular mechanisms involved in the expression of adipophilin in TNBC, and to develop therapeutic interventions for adipophilin-positive TNBC patients with a high risk of recurrence.

## Supporting information

**S1 Table. Clinical characteristics of patients with triple-negative breast cancer.**
(DOCX)

**S1 Fig. Flowchart of the exclusion criteria in this study.**
(TIFF)

**S1 Data.**
(PDF)

**S2 Data.**
(CSV)

## Acknowledgments

The authors would like to thank Editage for English language editing.

## Author Contributions

**Conceptualization:** Katsuhiro Yoshikawa, Mitsuaki Ishida.

**Data curation:** Katsuhiro Yoshikawa, Hirotsugu Yanai.

**Formal analysis:** Katsuhiro Yoshikawa, Mitsuaki Ishida.

**Investigation:** Katsuhiro Yoshikawa, Mitsuaki Ishida.

**Methodology:** Katsuhiro Yoshikawa.

**Project administration:** Katsuhiro Yoshikawa, Mitsuaki Ishida.

**Resources:** Katsuhiro Yoshikawa.

**Supervision:** Mitsuaki Ishida.

**Validation:** Mitsuaki Ishida.

**Writing – original draft:** Katsuhiro Yoshikawa.

**Writing – review & editing:** Katsuhiro Yoshikawa, Mitsuaki Ishida, Hirotsugu Yanai, Koji Tsuta, Mitsugu Sekimoto, Tomoharu Sugie.

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
