## [Decision Letter · Decision Letter 0]

7 Aug 2020

PONE-D-20-22115

Adipophilin expression is an independent marker for poor prognosis of patients with triple-negative breast cancer: An immunohistochemical study

PLOS ONE

Dear Dr. Mitsuaki Ishida,

Thank you for submitting your manuscript to PLOS ONE. After careful consideration, we feel that it has merit but does not fully meet PLOS ONE’s publication criteria as it currently stands. As you will see, the reviewers feel that significant additional work is necessary before your manuscript can be further considered for publication. Therefore, we invite you to submit a revised version of the manuscript that addresses the points raised during the review process.

Please submit your revised manuscript within three months. If you will need more time than this to complete your revisions, please reply to this message or contact the journal office at plosone@plos.org. Please include the following items when submitting your revised manuscript:

We look forward to receiving your revised manuscript.

Kind regards,

Elda Tagliabue

Academic Editor

PLOS ONE

Journal Requirements:

Reviewers' comments:

Reviewer's Responses to Questions

**Comments to the Author**

1. Is the manuscript technically sound, and do the data support the conclusions?

Reviewer #1: Yes

Reviewer #2: Partly

2. Has the statistical analysis been performed appropriately and rigorously? 

Reviewer #1: Yes

Reviewer #2: I Don't Know

3. Have the authors made all data underlying the findings in their manuscript fully available?

Reviewer #1: Yes

Reviewer #2: Yes

4. Is the manuscript presented in an intelligible fashion and written in standard English?

Reviewer #1: Yes

Reviewer #2: Yes

5. Review Comments to the Author

Reviewer #1: The manuscript by Yoshikawa et al. evaluated the prognostic value of adipophilin, as evaluated by IHC, in a cohort of 61 TNBC. Obtained results indicated that adipophilin-positive cases have worse prognosis than negative ones.

Overall this work is interesting and timing but there are some points that need the authors’ attention:

Major points:

a. The cut-off used (30% of positive cells) to define adipophilin positivity has never been used in other publications. The authors have to define why they chose this cut-off. A distribution of the % of positive cells in the cohort should be shown. Moreover, they have 23% positivity that is quite different from the 50% positivity in ER- tumors of Kuniyoshi et al paper. Discussion on this point has to be added.

b. Based on the year of patient recruitment (2006-2018), it would be interesting to extend survival analysis beyond 3 years of follow-up.

c. The authors have to demonstrate that adipophilin prognostic value in untreated TNBC patients is independent from other clinico-pathological features. Is the prognostic value of adipophilin independent from adjuvant chemotherapy treatment in multivariate analysis?

Minor points:

d. Please include details in Table S1 of the type of TNBC included in the analysis and specify among exclusion criteria in M&M the special types excluded.

e. Details on IHC method to evaluate adipophilin need to be included.

Reviewer #2: The authors studied expression of adipophilin (ADP) in 61 patients with triple-negative breast cancer (TNBC), and proposed that ADP may be a novel marker for the poor prognosis of patients with TNBC. This paper contains novel information and results which may be valuable for the readers. However, there are some major issues to be revised.

1) Authors describe that the tumor cells exhibited globular ADP staining pattern. Globular staining reminds the reviewer of sebaceous carcinoma. What is the criteria between globular and granular staining pattern?

2) Previously, ADP expression has been reported in apocrine carcinoma and lipid rich carcinoma of the breast (PMID: 21566511). Was their any morphologic characteristics in your ADP-positive breast carcinoma?

3) In the manuscript, the authors describe that "All patients were diagnosed with TNBC based on

108 biopsy results, and all samples were invasive carcinoma, no special type". Is their any possibility of discrepancy in the status between biopsy and surgically resected samples? Please refer to the current treatment guidlines.

4) Was histopathological grading performed on biopsy, surgical specimen, or TMA?

5) Since ADP staining was assessed on TMA, Ki67 index should be also analyzed on TMA instead of the preoperative biopsy specimens in order to investigate their relationship. Expression of ADP and Ki67 may be hetwrogeneous. Thus the sainings are better assessed at the same tumor site.

6) Regarding ADP expression and patients' prognosis, what is possible reasons for the discrepant results between TNBC with high and low Ki67 index? Again, ADP and Ki67 are better to be assessed on the same section. Study design should be reconsidered.

6. PLOS authors have the option to publish the peer review history of their article (what does this mean?). If published, this will include your full peer review and any attached files.

Reviewer #1: No

Reviewer #2: No

---

## [Author Response · Author response to Decision Letter 0]

8 Oct 2020

September 29, 2020

Dr. Joerg Heber

Editor-in-Chief

PLoS ONE

Resubmission - Manuscript ID: PONE-D-20-22115

Dear Dr. Heber:

I would like to resubmit an article for publication in PLoS ONE, titled “Adipophilin expression is an independent marker for poor prognosis of patients with triple-negative breast cancer: An immunohistochemical study”. 

Your comments, as well as those of the reviewers, were highly insightful and enabled us to greatly improve the quality of our manuscript. In the following pages, I have provided our point-by-point response to all comments and have highlighted the revised portions of the manuscript.

I hope that you will find our revised manuscript suitable for publication in PLoS ONE. I look forward to hearing from you at your earliest convenience.

Sincerely,

Mitsuaki Ishida 

Kansai Medical University 

2-5-1, Shinmachi, Hirakata City

Osaka, 573-1010, Japan

Tel: +81-72-804-2794

Fax: +81-72-804-2794

E-mail: ishidamt@hirakata.kmu.ac.jp

Response to the comments of Reviewer #1

Thank you very much for reviewing our manuscript. We appreciate your constructive comments. We have made the following revisions in response to the issues raised by you.

a. The cut-off used (30% of positive cells) to define adipophilin positivity has never been used in other publications. The authors have to define why they chose this cut-off. A distribution of the % of positive cells in the cohort should be shown. Moreover, they have 23% positivity that is quite different from the 50% positivity in ER- tumors of Kuniyoshi et al paper. Discussion on this point has to be added.

Response: Kaplan–Meier curves with cut-off values of 10, 20, 30, 40, and 50%, respectively, were generated and evaluated. As cut-off values of 30% significantly correlated with relapse-free survival, they were adopted in the present study. Kuniyoshi et al. showed that positive rate of adipophilin (perilipin 2) in basal type (=triple negative breast cancer) was 27% (cut-off value of 50%), and this corresponded with the results of the present study (23% of TNBC was adipophilin-positive in the present study).

“In order to determine the cut-off value for adipophilin positivity, analysis was performed with positive cut-off values of 10, 20, 30, 40, and 50%” (page 8, lines 108-110)

“Only cut-off value of 30% was significantly associated with relapse-free survival (p = 0.032), while cut-off values of 10%, 20%, 40%, and 50% were not significantly associated with relapse-free survival (p = 0.280, 0.072, 0.07, and 0.07, respectively). Therefore, in the present study, the cut-off value was set at 30% for the subsequent analyses” (page 9, lines 140-143)

b. Based on the year of patient recruitment (2006-2018), it would be interesting to extend survival analysis beyond 3 years of follow-up.

Response: As you pointed out, it would be interesting to analyze survival data beyond the 3-years of follow-up period. However, the number of patients whose clinical information was available for more than 3 years was not high in this present cohort. Thus, we did not show the survival data beyond 3 years of follow-up. 

c. The authors have to demonstrate that adipophilin prognostic value in untreated TNBC patients is independent from other clinico-pathological features. Is the prognostic value of adipophilin independent from adjuvant chemotherapy treatment in multivariate analysis?

Response: As you suggested, we analyzed adipophilin expression and adjuvant chemotherapy in relapse-free survival. Adipophilin expression was found to be independent from adjuvant chemotherapy (hazard ratio: 4.118; 95% confidence interval; 1.02–16.6; p = 0.047). We added this information in the manuscript as follows: 

“Moreover, adipophilin expression was independent from adjuvant chemotherapy in RFS (hazard ratio: 4.12; 95% confidence interval; 1.02–16.6; p = 0.047)” (page 14, lines 185-187) 

Minor point

d. Please include details in Table S1 of the type of TNBC included in the analysis and specify among exclusion criteria in M&M the special types excluded.

Response: Following your instructions, we have created a flowchart depicting the exclusion criteria with details of the histological type (Figure S1).

e. Details on IHC method to evaluate adipophilin need to be included.

Response: Based on your suggestion, the details of immunostaining were added.

“A primary mouse monoclonal antibody was used to detect adipophilin (AP125, Progen Biotechnik, Heidelberg, Germany; diluted at 1:100). For antigen retrieval, tissue sections were autoclaved at 100°C for 20 minutes in a tris-based buffer, pH 8.5, (Conditioning Solution CC1, Ventana Medical System). Thereafter, the automated protocol steps for immunostaining were followed. In addition, 3,3’-diaminobenzidine (DAB) was used as a colorimetric agent. Human sebaceous glands tissues were used as positive controls for adipophilin immunoreactivity” (page 7, lines 95-101)

Response to the comments of Reviewer #2

Thank you very much for reviewing our manuscript. We appreciate your constructive comments. We have made the following revisions in response to the issues raised by you.

1) Authors describe that the tumor cells exhibited globular ADP staining pattern. Globular staining reminds the reviewer of sebaceous carcinoma. What is the criteria between globular and granular staining pattern?

Response: The criteria for adipophilin staining patterns were based on reference 14.

Granular staining pattern is defined as staining of smaller subnuclear structure regions, while globular staining is defined as staining of larger pan-cytoplasmic regions. 

“Adipophilin expression was considered positive when the neoplastic cells showed granular and/or globular cytoplasmic expression as reported previously (Granular staining pattern is defined as staining of smaller subnuclear structure regions, while globular staining pattern is defined as staining of larger pan-cytoplasmic regions).” (page 7, lines 104-106)

2)Previously, ADP expression has been reported in apocrine carcinoma and lipid rich carcinoma of the breast (PMID: 21566511). Was their any morphologic characteristics in your ADP-positive breast carcinoma?

Response: This study did not contain any cases of apocrine carcinoma and lipid-rich carcinoma. Moreover, no differences were observed between the morphological features of adipophilin-positive and -negative cases. 

“Moreover, no differences were observed between the morphological features of adipophilin-positive and -negative cases” (page 10, lines 150-152)

3)In the manuscript, the authors describe that "All patients were diagnosed with TNBC based on biopsy results, and all samples were invasive carcinoma, no special type". Is their any possibility of discrepancy in the status between biopsy and surgically resected samples? Please refer to the current treatment guidlines.

Response: No discrepancy was found in pathological diagnosis and molecular subtype between pre-operative biopsy specimens and operative specimens in our cohort, which is similar to the previous reports that have shown high concordance between biopsy and surgically results, especially in patients with hormone-negative and HER2-negative cancer (Ann Oncol. 2009; 20(12):1948-52).

“No discrepancy was found in the pathological diagnosis and molecular subtype between the pre-operative biopsy specimens and operative specimens” (page 8, line 125 – Page 9, 127)

4)Was histopathological grading performed on biopsy, surgical specimen, or TMA?

Response: We evaluated the histopathological grading using TMA. I added some sentences in the manuscript to make it easier to understand.

“Histopathological grading was based on the Nottingham histological grade evaluated in the tumor tissue using tissue microarray [18]” (page 6, lines 79-81)

5)Since ADP staining was assessed on TMA, Ki67 index should be also analyzed on TMA instead of the preoperative biopsy specimens in order to investigate their relationship. Expression of ADP and Ki67 may be heterogeneous. Thus the stainingsare better assessed at the same tumor site.

Response: Following your recommendation, we re-analyzed all statistical data and determined Ki-67 LI of TMA; the statistical data associated with Ki-67 LI were all changed.

“The Ki-67 LI, which was also evaluated in the tissue microarray, was considered high when it was 40% or more, according to a meta-analysis of TNBC patients with tissue microarray [19]. The Ki-67 LI of two patients were not evaluated because less than 1,000 carcinoma cells were present in the tissue microarray” (page 6, lines 81-84)

“A high Ki-67 LI was significantly correlated with adipophilin expression (p = 0.005),” (page 10, lines 147-148)

6) Regarding ADP expression and patients' prognosis, what is possible reasons for the discrepant results between TNBC with high and low Ki67 index? Again, ADP and Ki67 are better to be assessed on the same section. Study design should be reconsidered.

Response: As you suggested, we assessed the immunohistochemical staining of Ki-67 LI and adipophilin on the same section using tissue microarray. All statistical analyses were re-performed using new Ki-67 LI data. However, the results remained unchanged; adipophilin expression was a significantly worse prognostic marker for patients with TNBC. 

As you mentioned, Ki-67 has been a well-known prognostic marker for patients with TNBC, and the present study clearly showed that adipophilin expression was a significantly worse prognostic marker and it might be superior to Ki-67 because among TNBC patients with a high Ki-67 LI, those negative for adipophilin exhibited better prognosis than those positive for adipophilin. Among adipophilin-negative patients, those with higher Ki-67 LI showed slightly better prognosis compared to those with lower Ki-67 LI (not significant). As you suggested, additional studies with larger number of patients are needed to clarify the significance of Ki-67 and adipophilin in prognosis of patients with TNBC.

---

## [Decision Letter · Decision Letter 1]

22 Oct 2020

PONE-D-20-22115R1

Adipophilin expression is an independent marker for poor prognosis of patients with triple-negative breast cancer: An immunohistochemical study

PLOS ONE

Dear Dr. Ishida,

Thank you for submitting your manuscript to PLOS ONE. After careful consideration of the amended version, there are still some issues that should be answered. Therefore, we invite you to submit a revised version of the manuscript that addresses the points raised during the review process.

We look forward to receiving your revised manuscript.

Kind regards,

Elda Tagliabue

Academic Editor

PLOS ONE

Reviewers' comments:

Reviewer's Responses to Questions

**Comments to the Author**

1. If the authors have adequately addressed your comments raised in a previous round of review and you feel that this manuscript is now acceptable for publication, you may indicate that here to bypass the “Comments to the Author” section, enter your conflict of interest statement in the “Confidential to Editor” section, and submit your "Accept" recommendation.

Reviewer #1: All comments have been addressed

Reviewer #2: All comments have been addressed

2. Is the manuscript technically sound, and do the data support the conclusions?

Reviewer #1: Yes

Reviewer #2: Partly

3. Has the statistical analysis been performed appropriately and rigorously? 

Reviewer #1: Yes

Reviewer #2: I Don't Know

4. Have the authors made all data underlying the findings in their manuscript fully available?

Reviewer #1: Yes

Reviewer #2: Yes

5. Is the manuscript presented in an intelligible fashion and written in standard English?

Reviewer #1: Yes

Reviewer #2: Yes

6. Review Comments to the Author

Reviewer #1: (No Response)

Reviewer #2: 1. Regarding the criteria for adipophilin staining patterns, reference 14 does not clearly define granular and globular staining patterns, but I presume that granular and globular staining patterns are defined by the size and shape of adipophilin positive substance instead of the stained region in the cells.

2. You mentioned that morphologic characteristics of ADP-positive and -negative breast carcinoma did not differ. Please specify the parameters you compared between the 2 groups; foamy cytoplasm, eosinophilic cytoplasm, prominent nucleoli, etc.

3. I would drop the following parts in the abstract and conclusion; "and might be superior to Ki-67 as a prognostic marker" and "Adipophilin may be a superior prognostic marker compared to Ki-67".

7. PLOS authors have the option to publish the peer review history of their article (what does this mean?). If published, this will include your full peer review and any attached files.

Reviewer #1: No

Reviewer #2: No

---

## [Author Response · Author response to Decision Letter 1]

3 Nov 2020

November 3, 2020

Dr. Joerg Heber

Editor-in-Chief

PLoS ONE

Resubmission - Manuscript ID: PONE-D-20-22115 R1

Dear Dr. Heber:

I would like to resubmit an article for publication in PLoS ONE, titled “Adipophilin expression is an independent marker for poor prognosis of patients with triple-negative breast cancer: An immunohistochemical study”. 

Your comments, as well as those of the reviewer, were highly insightful and enabled us to greatly improve the quality of our manuscript. In the following pages, I have provided our point-by-point response to all comments and have highlighted the revised portions of the manuscript.

I hope that you will find our revised manuscript suitable for publication in PLoS ONE. I look forward to hearing from you at your earliest convenience.

Sincerely,

Mitsuaki Ishida 

Kansai Medical University 

2-5-1, Shinmachi, Hirakata City

Osaka, 573-1010, Japan

Tel: +81-72-804-2794

Fax: +81-72-804-2794

E-mail: ishidamt@hirakata.kmu.ac.jp

Response to the comments of Reviewer #2

Thank you very much for reviewing our manuscript. We appreciate your constructive comments. We have made the following revisions in response to the issues raised by you.

1. Regarding the criteria for adipophilin staining patterns, reference 14 does not clearly define granular and globular staining patterns, but I presume that granular and globular staining patterns are defined by the size and shape of adipophilin positive substance instead of the stained region in the cells.

Response: Based on your suggestion, we have clarified the definition of adipophilin staining.

“Granular staining pattern is defined as staining of round smaller size, while globular staining pattern is defined as staining of round to oval larger size.” (page 8, line 109 - 111)

2. You mentioned that morphologic characteristics of ADP-positive and -negative breast carcinoma did not differ. Please specify the parameters you compared between the 2 groups; foamy cytoplasm, eosinophilic cytoplasm, prominent nucleoli, etc.

Response: Based on your suggestion, we evaluated the relationship between ADP and some morphologic features, including clear cytoplasm, eosinophilic cytoplasm, and prominent nucleoli. These features were not correlated with ADP expression. We added the results in Results and Table 1. 

“We also evaluated the presence of clear cytoplasm, eosinophilic cytoplasm, and prominent nucleoli in the tumor tissue using tissue microarray. Clear or eosinophilic cytoplasm was considered to be present when these histological changes were identified in >5% of the tumour cells, as described earlier [14].” (page 6, line 82 – 85)

“Moreover, no differences were observed between the morphological features of adipophilin-positive and -negative cases, including the presence of clear cytoplasm, eosinophilic cytoplasm, and prominent nucleoli.” (page 10, line 155 - 158)

3. I would drop the following parts in the abstract and conclusion; "and might be superior to Ki-67 as a prognostic marker" and "Adipophilin may be a superior prognostic marker compared to Ki-67".

Response: Based on your suggestion, we have dropped the specified part in the Abstract and Conclusion.

---

## [Decision Letter · Decision Letter 2]

5 Nov 2020

Adipophilin expression is an independent marker for poor prognosis of patients with triple-negative breast cancer: An immunohistochemical study

PONE-D-20-22115R2

Dear Dr. Ishida,

We’re pleased to inform you that your manuscript has been judged scientifically suitable for publication and will be formally accepted for publication once it meets all outstanding technical requirements.

Kind regards,

Elda Tagliabue

Academic Editor

PLOS ONE

Additional Editor Comments (optional):

Reviewers' comments:

Reviewer's Responses to Questions

**Comments to the Author**

1. If the authors have adequately addressed your comments raised in a previous round of review and you feel that this manuscript is now acceptable for publication, you may indicate that here to bypass the “Comments to the Author” section, enter your conflict of interest statement in the “Confidential to Editor” section, and submit your "Accept" recommendation.

Reviewer #2: All comments have been addressed

2. Is the manuscript technically sound, and do the data support the conclusions?

Reviewer #2: Yes

3. Has the statistical analysis been performed appropriately and rigorously? 

Reviewer #2: I Don't Know

4. Have the authors made all data underlying the findings in their manuscript fully available?

Reviewer #2: Yes

5. Is the manuscript presented in an intelligible fashion and written in standard English?

Reviewer #2: Yes

6. Review Comments to the Author

Reviewer #2: (No Response)

7. PLOS authors have the option to publish the peer review history of their article (what does this mean?). If published, this will include your full peer review and any attached files.

Reviewer #2: No

---

## [Editor Report · Acceptance letter]

9 Nov 2020

PONE-D-20-22115R2 

Adipophilin expression is an independent marker for poor prognosis of patients with triple-negative breast cancer: An immunohistochemical study 

Dear Dr. Ishida:

I'm pleased to inform you that your manuscript has been deemed suitable for publication in PLOS ONE. Congratulations! Your manuscript is now with our production department. 

Kind regards, 

on behalf of

Dr. Elda Tagliabue 

Academic Editor

PLOS ONE